# Neutrophils Modulate Fibroblast Function and Promote Healing and Scar Formation after Murine Myocardial Infarction [note 1]

**DOI:** 10.3390/ijms21103685

**Published:** 2020-05-23

**Authors:** Adelina Curaj, David Schumacher, Mihaela Rusu, Mareike Staudt, Xiaofeng Li, Sakine Simsekyilmaz, Vera Jankowski, Joachim Jankowski, Andreea Ramona Dumitraşcu, Derek J Hausenloy, Alexander Schuh, Elisa A. Liehn

**Affiliations:** 1Institute for Molecular Cardiovascular Research (IMCAR), RWTH Aachen University, 52074 Aachen, Germany; acuraj@ukaachen.de (A.C.); dschumacher@ukaachen.de (D.S.); mrusu@ukaachen.de (M.R.); mstaudt@ukaachen.de (M.S.); xiaofenglibio@gmail.com (X.L.); sakine.simsekyilmaz@uk-koeln.de (S.S.); vjankowski@ukaachen.de (V.J.); jjankowski@ukaachen.de (J.J.); dumitrascu.andre@yahoo.com (A.R.D.); 2Victor Babes National Institute of Pathology, 050096 Bucharest, Romania; 3Department of Anesthesiology, University Hospital, RWTH Aachen University, 52074 Aachen, Germany; 4Institut für Pharmakologie und Klinische Pharmakologie, University Hospital, 40225 Düsseldorf, Germany; 5Experimental Vascular Pathology, Cardiovascular Research Institute Maastricht (CARIM), University of Maastricht, 6200 Maastricht, The Netherlands; 6Human Genetic Laboratory, University of Medicine and Pharmacy, 200642 Craiova, Romania; 7Cardiovascular & Metabolic Disorders Program, Duke-National University of Singapore Medical School, Singapore 169857, Singapore; d.hausenloy@ucl.ac.uk; 8National Heart Research Institute Singapore, National Heart Centre, Singapore 169609, Singapore; 9Yong Loo Lin School of Medicine, National University Singapore, Singapore 169857, Singapore; 10The Hatter Cardiovascular Institute, University College London, London WC1E 6BT, UK; 11Cardiovascular Research Center, College of Medical and Health Sciences, Asia University, Taichung 41354, Taiwan; 12Department of Cardiology, Pulmonology, Angiology and Intensive Care, University Hospital, RWTH Aachen University, 52074 Aachen, Germany; aschuh@ukaachen.de

**Keywords:** myocardial infarction, inflammation, neutrophils, fibroblasts, extracellular matrix formation, scar formation

## Abstract

Aim: Recruitment of neutrophils to the heart following acute myocardial infarction (MI) initiates inflammation and contributes to adverse post-infarct left ventricular (LV) remodeling. However, therapeutic inhibition of neutrophil recruitment into the infarct zone has not been beneficial in MI patients, suggesting a possible dual role for neutrophils in inflammation and repair following MI. Here, we investigate the effect of neutrophils on cardiac fibroblast function following MI. Methods and Results: We found that co-incubating neutrophils with isolated cardiac fibroblasts enhanced the production of provisional extracellular matrix proteins and reduced collagen synthesis when compared to control or co-incubation with mononuclear cells. Furthermore, we showed that neutrophils are required to induce the transient up-regulation of transforming growth factor (TGF)-ß1 expression in fibroblasts, a key requirement for terminating the pro-inflammatory phase and allowing the reparatory phase to form a mature scar after MI. Conclusion: Neutrophils are essential for both initiation and termination of inflammatory events that control and modulate the healing process after MI. Therefore, one should exercise caution when testing therapeutic strategies to inhibit neutrophil recruitment into the infarct zone in MI patients.

## 1. Introduction

Following myocardial infarction (MI), a number of complex inflammatory and reparatory processes are initiated, which act to remove and replace necrotic heart tissue. A persistent pro-inflammatory response can impair healing and contribute to adverse left ventricular (LV) remodeling after MI [1]. Neutrophils are recruited into the infarct zone immediately after MI onset, and their rapid degranulation and degradation have been shown to play a major pro-inflammatory role, contributing to an increase in MI size by inducing the death of cardiomyocytes at the border zone (so-called “neutrophil-induced injury”) [2], and impairing wound healing, resulting in adverse post-infarct LV modeling [3,4,5].

Despite encouraging results in experimental animal models [6,7,8,9], therapeutic targeting of neutrophils following MI has had mixed results [8,10,11,12,13,14]. The reduction of neutrophil infiltration into the infarct zone has been reported in some studies to reduce MI size and preserve heart function after MI [8], whereas other studies have revealed no beneficial effects [11,12] or have even documented a deterioration in heart function after MI [10,15,16]. Moreover, the persistence of neutrophils in the infarcted area has been reported to influence neither heart function nor infarct size [17]. Therefore, there is a need to elucidate the role of neutrophils in healing and scar formation after MI in order to design effective therapeutic strategies for preventing adverse LV remodeling.

It is already known that immune cells recruited into the infarct zone after MI can interact with cardiac fibroblasts, and that this is required for extracellular matrix synthesis and scar formation [18]. Cardiac fibroblasts are directly activated after MI, and during their proliferation, they produce an initial provisional extracellular matrix, rich in fibrin/fibronectin [19]. Later, their function switches to synthesize the collagen-based mature scar [19]. To date, the cellular and molecular mechanisms modulating fibrosis post-MI are not known. We hypothesize that immune cells, especially neutrophils, play the main role in regulating and controlling fibrosis following MI.

## 2. Results

### 2.1. Neutrophil Characterization

The purity of the neutrophils was checked using FACS staining for Ly-6G, CD115 and F4/80 for up to four days in culture. The neutrophil fraction was pure and did not stain for monocytic CD115 (Figure 1A) or macrophages markers F4/80 (Figure 1B). The viability of the neutrophils was not affected during the culturing (Figure 1C). The mononuclear fraction was positive for monocytic markers CD115 (Figure 1D), showing differentiation after 4 days in culture (F4/80 positive). Based on the stability of the cells after 1 day in culture, we performed further co-culture experiments using 24 h incubation periods. 

### 2.2. Neutrophil-Mediated Changes in Extracellular Matrix Protein Synthesis

Cardiac fibroblasts were isolated and cultured in the presence of blood neutrophil fraction (Ne) in hypoxia (H), hence mimicking the in vivo conditions during and after MI. We demonstrated that neutrophils and neutrophil-secreted mediators (neutrophil fraction) induced a significant increase in mRNA expression of fibronectin (Figure 2A) in isolated cardiac fibroblasts. Simultaneously, the neutrophil fraction induced a significant decrease in mRNA expression of collagen I (Figure 2B), when compared to fibroblasts co-incubated with mononuclear fraction, features which are characteristic of the provisional extracellular matrix. These results were validated by MALDI-mass fingerprint-spectra of tryptic peptides of these proteins. Fibronectin (Figure 2C) protein expression was detected only in fibroblasts co-incubated with neutrophil fraction. Similarly, collagen was detected only in fibroblasts co-incubated with mononuclear fraction (Figure 2D). Neutrophils and mononuclear cells had different effects regarding other inflammatory genes: While neutrophils activated the IL-1β (Figure 3E), mononuclear fraction activated PPAR γ (Figure 3F).

### 2.3. Treatment Efficiency in Animal Model 

The efficiency of the treatment was checked by FACS from blood samples one day after initiation of the treatment (Figure 3A,B). As expected, we observed a drastic decrease in neutrophil numbers, but also in inflammatory monocytes. These findings are consistent with those from Horckmans et al. [10], who demonstrated that despite impaired recruitment of these cells, the content of macrophages was not affected. It was even increased at later time points after MI. However, neutrophils were drastically reduced in the heart one day after MI (Figure 3C), while macrophages showed no visible changes at later time points after MI (Figure 3D).

### 2.4. Neutrophil-Mediated Changes in TGF-β1 Expression

Neutrophils were also found to increase mRNA (Figure 4A) and protein expression (Figure 4B) levels of transforming growth factor (TGF)-ß1 in isolated fibroblasts under hypoxic conditions. Interestingly, TGF-β1 was found to be highly expressed in fibroblasts (–TGF-β1 stimulation), but not in differentiated myofibroblasts (+TGF-β1 stimulation, Figure 4A), suggesting a negative feedback loop of TGF-β1 -regulation at elevated concentrations. This suggests a biphasic influence of neutrophils on fibroblasts: (1) Directly after MI, neutrophils induce increased TGF-β1 production in fibroblasts. TGF-β1 helps to switch the pro-inflammatory towards anti-inflammatory processes. After neutrophil depletion, the fibroblasts do not produce TGF-β1. The switch to anti-inflammatory processes is delayed. (2) When fibroblasts were differentiated towards myofibroblasts, TGF-β1 production decreased significantly. This was independent of neutrophil depletion or non-depletion.

These results were confirmed by the time course of TGF-β1 mRNA expression in myocardium after MI (Figure 4C, black columns). After short-term down-regulation, presumably due to tissue necrosis, TGF-β1 increased significantly at one and two weeks after MI and decreased rapidly thereafter. Double immunofluorescence staining co-localized TGF-β1 expression in fibroblasts at one and two weeks after MI (Figure 4D, right panels). To demonstrate the role of neutrophils in up-regulating the TGF-β1 expression in fibroblasts, neutrophil depletion was performed in vivo. In the absence of neutrophils, TGF-β1 expression was not increased in the infarcted areas (Figure 4D, left panels) and in fibroblasts (Figure 4D, left panels, inset).

Since TGF-β1 is essential for resolution of the pro-inflammatory phase following MI, we assessed how TGF-β1 expression in neutrophil-depleted mice influenced the inflammatory processes. Using the Interleukin (IL)-6 as a common inflammatory marker, we showed that while IL-6 was expressed at the earlier but not later time points after MI in control mice, it remained at significantly higher levels four weeks after MI in neutrophil-depleted mice (Figure 4E,F).

## 3. Discussion

The negative role of neutrophils in adverse post-infarct LV remodeling has been demonstrated by many studies [20]. However, neutrophils appear to play a double-edged role in healing after MI. The positive effects of neutrophils may explain, in part, the controversial results obtained after manipulating neutrophils in different experimental settings. Compared with the study of Horckmans et al., who demonstrated an important role of neutrophils in polarization of macrophages towards the reparatory phenotype [10], we have demonstrated in this study that neutrophils are actively involved in extracellular matrix formation, representing the key player in switching from the pro-inflammatory to anti-inflammatory phase after MI.

In the initial phase of scar formation, fibronectin is essential to stabilize the extracellular matrix [21], building a scaffold for later deposition of collagen type I [22]. Our data revealed that neutrophils regulated the phenotype of fibroblasts, promoting provisional matrix synthesis and delaying mature scar formation. This might explain the excessive fibrosis and increased collagen content induced by the depletion of neutrophils [10], resulting in impaired LV remodeling and heart failure [10] found by Horckmans et al. A possible mechanism can be represented by the neutrophil-dependent up-regulation of IL-1ß, which is demonstrated to decrease collagen synthesis and increase matrix metalloproteinase activity in cardiac fibroblasts [23]. IL-1ß is responsible for activation of the inflammasome in cardiac fibroblasts [24,25], triggering an inflammatory cascade, thus increasing fibroblast migration [26] and stimulating a matrix-degrading program [23].

Further, we found another important role of neutrophils, in modulating TGF-β1 expression in fibroblasts. TGF-β1 is a cytokine with diverse and ambiguous cellular effects, which is still far from being fully understood [27]. In the context of pressure overloaded myocardium, activation of TGF-β1 signaling is responsible for cardiomyocyte hypertrophy and excessive interstitial fibrosis [28]. Thus, targeting TGF-β1 is associated with markedly reduced collagen deposition, resulting in increased left ventricular function [29]. During healing after MI, TGF-β1 represents the major factor which determines the resolution of inflammation, allowing adequate healing [28,30]. In vitro, neutrophils are able to induce a rapid up-regulation of TGF-β1 in fibroblasts. In vivo, we observed a later up-regulation of TGF-β1 in myocardium, when the neutrophils are almost absent (Figure 4C,D). However, depleting neutrophils in these mice abolished the short-term up-regulation of TGF-β1. Moreover, since TGF-β1 is able to suppress the inflammatory cytokines, such as IL-6 [26], neutrophil depletion induced the persistence of these inflammatory cytokines over the analyzed period compared to the control (Figure 4E,F). This suggests that the effect of neutrophils on TGF-β1 up-regulation in fibroblasts after MI is more complex and mediated through other players, such as monocytes/macrophages [10].

Interestingly, TGF-β1 up-regulation was completely abolished in myofibroblasts, independent of immune cells (Figure 4A), suggesting a self-regulatory negative feedback of TGF-β1 expression in these cells (Figure 4A). This is the most important mechanism in the maturation of the scar and preservation of heart function, since persistence of TGF-β1 expression worsens post-infarct left ventricular remodeling [29,31].

While these effects are known to be mediated by Smad3 [32], the mechanism through which the latter inhibits the inflammation induced by TGF-β1 is unknown. We speculate that an interaction between TGF-β1 and polarized monocytes/macrophages switches the fibroblast phenotype, inhibiting their proliferation, stimulating their apoptosis, and increasing collagen synthesis by activation of PPAR-δ [33] and angiotensin II [34,35,36,37,38]. Thus, after fulfilling the mature scar, fibroblasts become inactive again, stabilizing the scar and completing the healing process after MI. Further studies are needed to elucidate the underlying mechanisms.

Study limitations: According to the current European regulations regarding animal experiments, we included at least five animals per group, while maintaining the minimum number of animals required for statistical analysis. Furthermore, we did not repeat long-term functional experiments [10], as we have focused on novel mechanistic insights, in order to further understand healing after MI.

In conclusion, our study has demonstrated that neutrophils play a key role in both induction and resolution of inflammation by modulating cardiac fibroblast function (Figure 5). Novel therapeutic strategies may consider selectively modulating the inflammatory processes, thereby allowing neutrophils to fulfil their modulatory function on cardiac fibroblast function and scar formation. Nevertheless, our data can be considered in a multi- and inter-disciplinary context, given that immune cells and scar formation are ubiquitously involved not only in the physiological homeostasis, but also in organ recovery and function preservation after injury.

## 4. Material and Methods

### 4.1. Cell Isolation and Co-Culture

Fibroblasts from adult murine hearts were isolated by enzymatic digestion (Liberase Blendzyme 1; Roche; Basel, Switzerland). The isolation procedures were in accordance with European legislation and approved by local German authorities (AZ: 8.87-50.10.35.09.088). The cell suspension was filtered using a 100 µm cell strainer (BD Falcon, Heidelberg, Germany). Cells obtained after centrifugation (400 g, 20 °C, 5 min) were plated on petri dishes for one hour to select the fibroblasts, then the adherent cells were cultured further in DMEM High Glucose (4,6 g/L) with L-Glutamine (PAA—The Cell Culture Company, Cölbe, Germany), 10% FBS (Fetal bovine serum dialyzed; PAN Biotech, Aidenbach, Germany) supplemented with 1% Penicillin/Streptomycin (PAA–The Cell Culture Company, Cölbe, Germany). After reaching confluence, the fibroblasts were cultured in normoxia (N) or hypoxia (H, 5% CO_2_, 2% O_2_, 37 °C, Innova^@^ CO-48 Incubator, New Brunswick Scientific, Enfield, CT, USA) to mimic in vivo conditions.

Neutrophils (+Ne) were isolated from mouse bone marrow using Hystopaque-1119 (Merck, Darmstadt, Germany) as described by the manufacturer. As control, we used the mononuclear fraction (+Mo) isolated from the same gradient centrifugation and maintained in the same conditions in all experiments. The purity of the neutrophils and mononuclear fractions was checked using FACS staining for Ly-6G, CD115 and F4/80 (eBioscience, Thermo Fisher Scientific, Schwerte, Germany) for up to four days in culture. Viability was tested using propidium iodide (Thermo Fisher Scientific, Schwerte, Germany). Due to the observed phenotype shift during the culture of neutrophils, further co-culture experiments with fibroblasts/myofibroblasts (ratio 1:1) were performed for 24 h.

As required, fibroblast differentiation into myofibroblast was induced with 100 ηg/mL TGF-β1 (PeproTech, Hamburg, Germany). Myofibroblast differentiation was confirmed by means of smooth muscle actin (SMA) staining in cultured cells and FACS for SMA expression. All cells were free of mycoplasma as determined by PCR during the regulatory laboratory check.

### 4.2. Animal Model of Myocardial Infarction

Nine-to-twelve-week-old male C57Bl/6 wild-type mice (Charles River, Köln, Germany) were subjected to chronic myocardial infarction (MI), as previously described [8,11]. Briefly, mice (*n* = 6) were intubated under general anesthesia using 100 mg/kg ketamine, 10 mg/kg xylazine, i.p. and ventilated with oxygen using a mouse respirator (Harvard Apparatus, March-Hugstetten, Germany). After exposing the hearts by left thoracotomy, MI was induced by occlusion of the left anterior descending artery (LAD) with 0/7 silk. The ribs, muscle layer, and skin incision were closed, and 0.1 mg/kg buprenorphin was administrated for the next days until full recovery. For neutrophil depletion, male C57Bl/6 wild-type mice (Charles River, Köln, Germany) were treated intraperitoneally with monoclonal antibody against Ly-6G, clone 1A8 (200 µg; BioXCell, Lebanon, NH, USA) [10]. To assure complete neutrophil depletion at the moment of vessel ligature, the treatment was administered 24 h before induction of MI, and then every day until the endpoint of each experiment.

According to the current European regulations regarding animal experiments, we aimed to include at least 5 animals per group, while maintaining the minimum number of animals required for statistical analysis.

The hearts were excised at predefined time points (at 0, 1, 4, 7, 14, 21, and 28 days) and prepared for further analysis. All mice were housed under standardized conditions in the Animal Facility of the University Hospital Aachen (Germany). The operating procedure was in accordance with European legislation and approved by local German authorities (AZ: 8.87-50.10.35.09.088). Mice were not excluded from the analysis unless they died during the open-chest operation. Since it has already been demonstrated that neutrophil depletion worsens heart function and increases infarct size [10], we did not repeat these experiments and focused on the role of neutrophils in TGF-β1 production from fibroblasts/myofibroblasts.

### 4.3. mRNA Isolation and RT-PCR

mRNA was isolated 24 h after culture in the indicated experimental conditions, using RNeasy Mini Kit (Qiagen, The Netherlands) after removing the co-cultured cells by repeated washing under the microscope. Each experiment was performed in three replicates and repeated 3 times on different days. Pooled results from all 3 days are presented (*n* = 9).

mRNA was isolated from mouse infarct zones using RNeasy Mini Kit (Qiagen, Venlo, The Netherlands) after tissue lysis and centrifuged through a QIAshredder homogenizer. Quantitative determination of extracellular matrix proteins was determined using specific primers (Table 1) and murine ß-actin as housekeeping gene. Each experiment was performed in three replicates and repeated 3 times on different days. Pooled results from all 3 days are presented (*n* = 9).

### 4.4. Mass Spectrometry Protein Analysis

Protein analysis was performed 24 h after culture in the indicated experimental conditions. To analyze protein expression, cell lysates (*n* = 3) were washed/equilibrated with ammonium bicarbonate in acetonitrile and digested with 0.02 µg trypsin at 37 °C for 24 h. The resulting peptides were desalted and concentrated using ZipTip_C18_ (Millipore, Burlington, MA, USA) technology. The eluates of the ZipTip_C18_ were spotted directly onto the matrix-assisted laser desorption/ionization (MALDI) target (Bruker-Daltonic, Bremen, Germany) using a-cyano-4-hydroxycinnamic acid as matrix. The subsequent analyses were carried out using a matrix-assisted laser desorption/ionization time-of-flight mass spectrometry (MALDI-TOF-MS) using MALDI-Lift fragment option (MALDI-TOF/TOF-MS). Calibrated and annotated spectra were subjected to a database search (Swiss-Prot, Zürich, Switzerland) using Bruker Bio Tool 3.2 and the Mascot 2.2 search engine, which compared the experimental MALDI-TOF-MS and MALDI-TOF/TOF-MS data sets with the calculated peptide masses in the sequence database for each entry. Using empirically determined factors, a statistical weight was assigned to each individual peptide match.

### 4.5. Immunofluorescence

Three sections per mouse (*n* = 5–6) were stained using anti- TGF-β1 antibody (Abcam, Cambridge, UK), anti-IL-6 antibody or anti-Mac3 (BD Pharmingen, San Jose, CA, USA) followed by fluorescein isothiocyanate (FITC)-conjugated secondary antibody. Double staining was performed using anti-smooth muscle actin (SMA, DAKO, Hamburg, Germany) or anti-MPO antibody (Neomarkers, Thermo Fisher Scientifics, Schwerte, Germany), followed by Cy3-conjugated secondary antibody (DAKO, Hamburg, Germany). The images were made using DISKUS (Hilgers, Königswinter, Germany). The contrast amplification and overlay were performed using DISKUS (Hilgers, Königswinter, Germany).

### 4.6. Statistical Analysis

Statistical analysis was performed with Prism5 software (GraphPad Software, San Diego, CA, USA) using 1-way ANOVA followed by Newman–Keuls post-hoc test, or 2-way ANOVA followed by Bonferroni test, as indicated. Data are presented as mean ± SEM values. *p*-values of <0.05 were considered significant.

### 4.7. Data Sharing Statement

For original data, please contact the corresponding author (eliehn@ukaachen.de).

## 5. Conclusions

Neutrophils stabilize the extracellular matrix by sustaining the provisional matrix protein synthesis.

Neutrophils are responsible for the short-term up-regulation of TGF-β1, and termination of inflammatory phase after myocardial infarction.

## Figures and Tables

**Figure 1 ijms-21-03685-f001:**
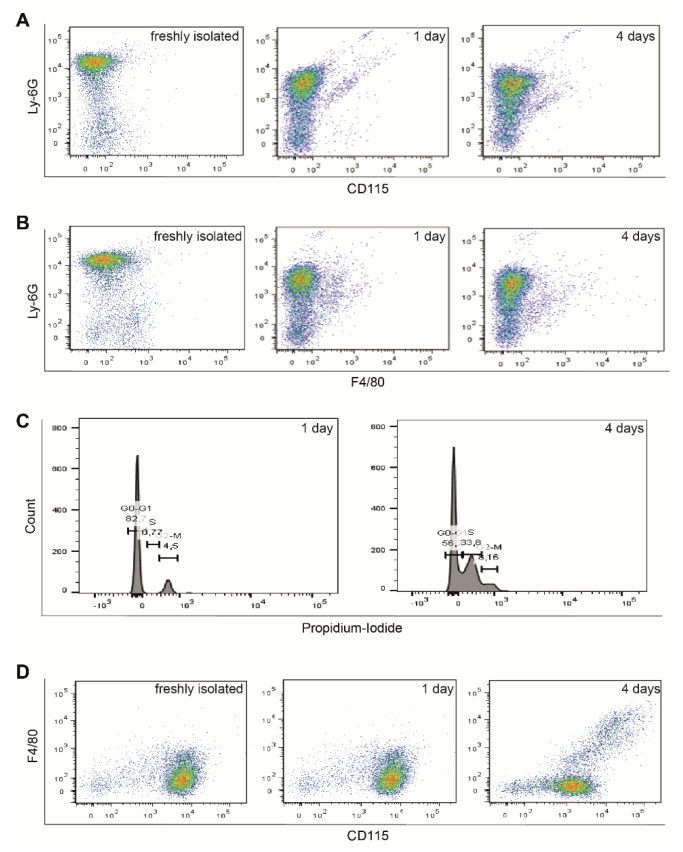
Validation of neutrophil and mononuclear fractions after isolation. The neutrophil fraction was positive for Ly-6G, but not for CD115 or F4/80 (**A**). Their phenotype did not change but was slightly shifted after 4 days in culture (**B**). The viability of the cells was not affected during culturing (**C**). Mononuclear fraction was positive for CD115 and showed differentiation potential towards macrophage after 4 days in culture (**D**).

**Figure 2 ijms-21-03685-f002:**
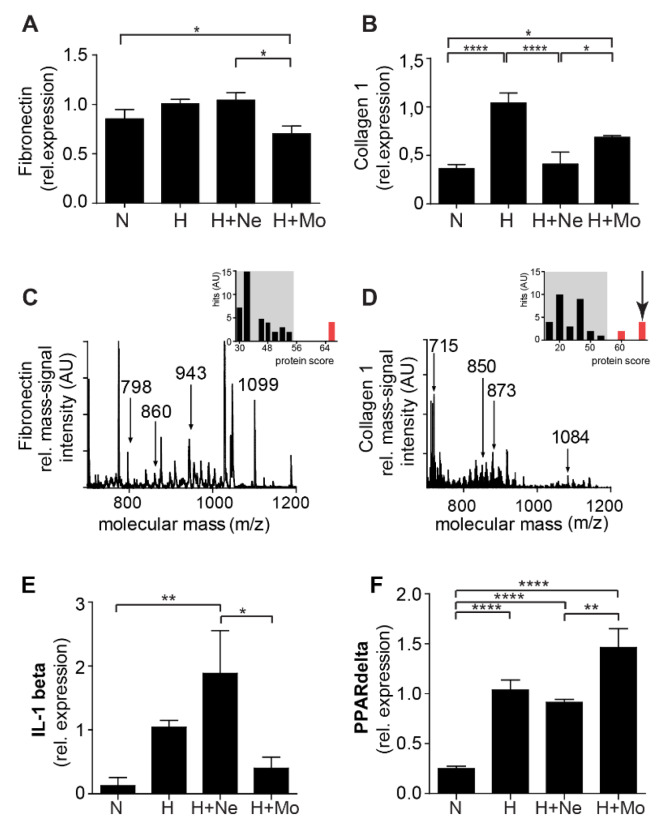
The effect of neutrophil-fraction on extracellular matrix protein synthesis. (**A**) Fibronectin mRNA expression and (**B**) collagen mRNA expression in isolated fibroblasts cultured in normoxia (N), hypoxia (H) and co-cultured with neutrophil (Ne) fraction (H+Ne) and mononuclear (Mo) fraction (H+Mo), respectively (*n* = 9, * *p* < 0.05, **** *p* < 0.0001). (**C**) Characteristic matrix-assisted laser desorption/ionization (MALDI) mass fingerprint-spectrum of tryptic peptides of fibronectin (*n* = 3). (**D**) Characteristic MALDI mass fingerprint-spectrum of tryptic peptides of collagen I detected in isolated fibroblasts co-incubated with mononuclear fraction (*n* = 3). Protein scores from the Mascot database are shown for each identification in right insets (arrow indicates the identified protein). Neutrophils (H+Ne) increased the gene expression of (**E**) IL-1ß (*n* = 9), while mononuclear cells (H+Mo) increased the gene expression of (**F**) PPAR γ (*n* = 9). (* *p* < 0.05; ** *p* < 0.01; **** *p* < 0.0001).

**Figure 3 ijms-21-03685-f003:**
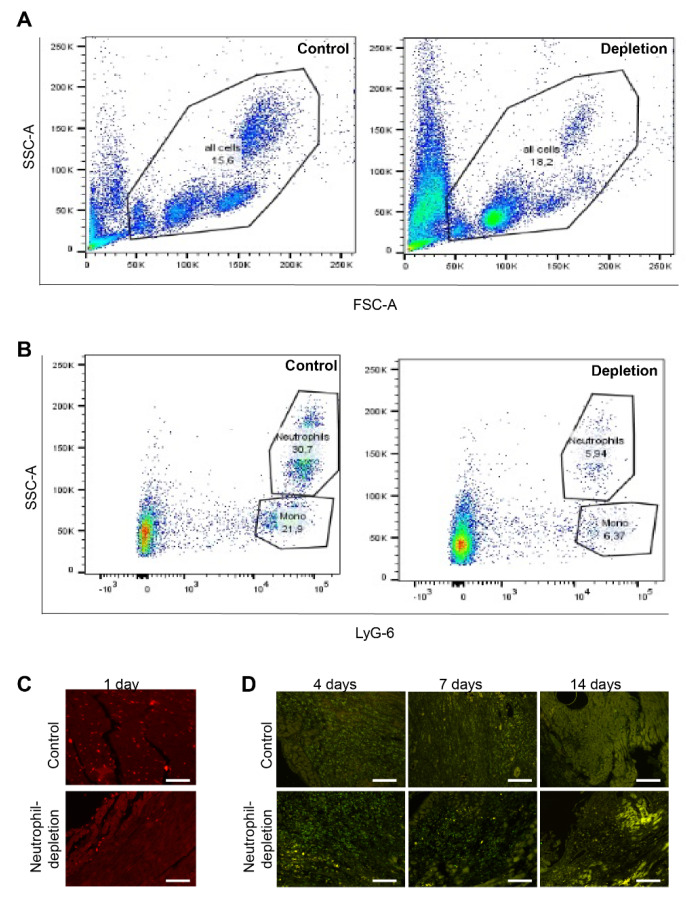
Validation of neutrophil-depletion treatment. Mice undergoing neutrophil depletion showed a significant reduction in blood neutrophils (**A**) and some inflammatory monocytes (**B**). However, while neutrophil infiltration was significantly reduced one day after myocardial infarction (MI) (**C**) (scale bar 50 µm), the macrophages were not affected by the treatment at later time points after MI (**D**) (scale bar 100 µm).

**Figure 4 ijms-21-03685-f004:**
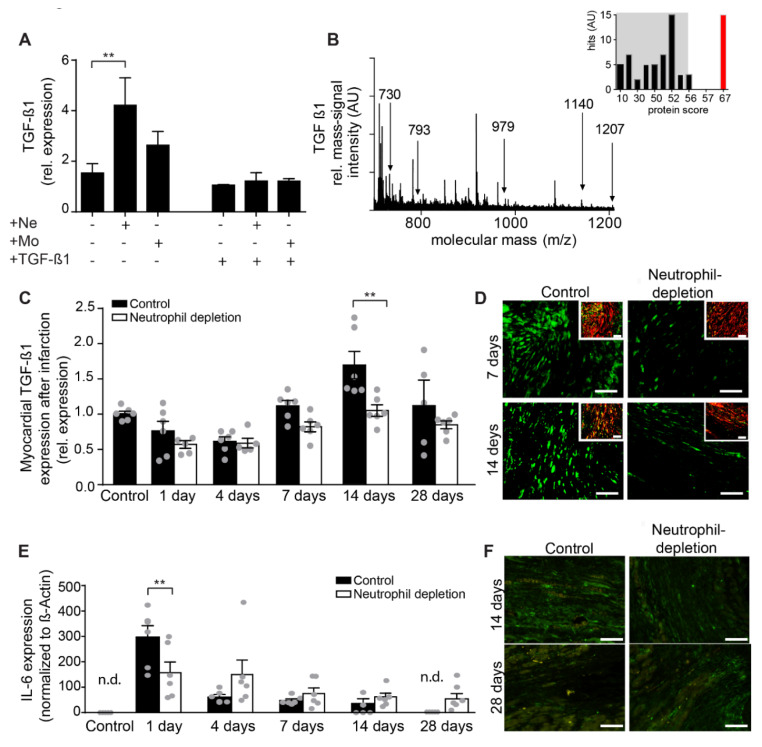
The effect of neutrophils on TGF-β1 dynamics during MI. (**A**) TGF-β1 mRNA expression in fibroblasts (–TGF-β1) and myofibroblasts (+TGF-β1) co-incubated under hypoxic conditions without/with neutrophil and mononuclear fractions, respectively (*n* = 4–8, ** *p* < 0.01). (**B**) Characteristic MALDI mass fingerprint-spectrum of tryptic peptides of TGF-β1 in fibroblast lysates after co-incubation with neutrophil fraction. Protein score from MASCOT database is shown in the right inset. (**C**) Time-dependent myocardial mRNA expression of TGF-β1 after MI (*n* = 5−6, ** *p* < 0.01). (**D**) TGF-β1 staining in myocardium by immunofluorescence (green) at different MI set points in control and in neutrophil-depleted mice (*n* = 5−6). Double immunofluorescence of TGF-β1 (green), smooth alpha actin (red) and overlay (yellow) at different MI set points is shown in insets (scale bar 50 µm). (**E**) Time-dependent myocardial mRNA expression of IL-6 after MI (*n* = 6, ** *p* < 0.01, *n.d.* not detected) in mice without (black columns) and with (white columns) neutrophil depletion. (**F**) Representative double immunofluorescence of IL-6 (green) at different MI set points is shown in insets (scale bar 50 µm).

**Figure 5 ijms-21-03685-f005:**
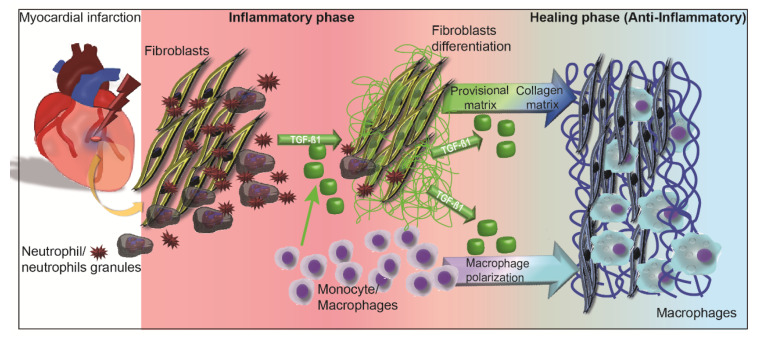
Sketch of healing after MI based on neutrophil-modulated fibroblast function. MI is followed by complex cellular and molecular events, in which fibroblasts proliferate, producing an initial provisional matrix (green fibers). In the inflammatory phase (red panel), the recruited neutrophils assist fibroblasts with provisional fibrin/fibronectin matrix formation (green fibers). TGF-β1 (green capsules) expressed from monocytes and neutrophil-stimulated fibroblasts leads to fibroblast differentiation, collagen synthesis (blue fibers) and macrophage polarization, respectively. Reaching an elevated expression, a self-regulatory negative feedback is activated, reducing TGF-β1 expression, allowing the healing phase to complete the mature scar.

**Table 1 ijms-21-03685-t001:** Primers used to determine mRNA expression.

ECM Protein	Forward Primer	Reverse Primer
Fibronectin	GTGACACCCACCAGCTTTAC	ATCACCGATGAGCTGTCTGG
Collagen I	ACTACTGGAGAAGTTGGCAAGC	GTACCACGTTCTCCTCTTGGAC
TGF-β1	AGTGTGGAGCAACATGTGAAC	TTCAGCCACTGCCGTACAAC
β-actin	AGCCATGTACGTAGCCATCC	CTCTCAGCTGTGGTGGTGAA
IL-1ß	CAACCAACAAGTGATATTCTCCA	GATCCACACTCTCCAGCTGCA
PPARδ	GGGGGTCAGTCATGGAACAG	GTGTGTTCTGGTCCCCCATT
IL-6	TCTGGAGTACCATAGCTACCTGGAGT	AGCATTGGAAATTGGGGTAGGAAGGA
TNF-α	GTCCCCAAAGGGATGAGAAG	AGATGATCTGAGTGTGAGGG

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
