# Peer review of "Neutrophils Modulate Fibroblast Function and Promote Healing and Scar Formation after Murine Myocardial Infarction †"

_ijms, 2020, doi:10.3390/ijms21103685_

Round 1
Reviewer 1 Report
Myocardial infarction is the most severe complication in coronary heart disease. The ischemia and subsequent reperfusion induce an innate immune response. It has been widely accepted that neutrophil induced innate immune response is closely related to fibroblast activation and polarization. However, a complicated scenario exists that inhibition of neutrophil infiltration protects the heart from acute ischemic insult but exacerbates fibrosis and heart failure. Here, Curaj et al. showed that neutrophil-fibroblast communication was important for post-infarct fibrosis via TGF-β1. There are several points to be addressed:
- To address the distinct effects of neutrophils at the acute or chronic stage of myocardial infarction, depletion prior to and after myocardial infarction is highly recommended.
- Neutrophil depletion by antibody 1A8 didn't generate a comprehensive result. Instead, both neutrophils and monocytes showed a remarkable reduction in peripheral blood as shown in Figure 2B. Then, one of the key findings as shown in Figure 4C or 4E cannot be easily explained by neutrophil depletion.
- Neutrophils, utilized in this study, were isolated from bone marrow. Based on the mechanisms of neutrophil activation, the experiments cannot recapitulate the situation in vivo, especially after myocardial infarction. Moreover, in vitro culturing affects immune cell phenotype. Figure 1 A or B clearly show such phenotype shift during culturing. The authors should further address this issue.
- Monocyte-derived macrophages or cardiac resident macrophages are the important innate immune players. There is a state change from monocytes in the bone marrow to macrophage. Thus, monocytes need further activation before the coculture experiments.
- The results from Figure 4A are inadequate to demonstrate the effects of neutrophil-fibroblast communication on TGF-β1 production in myocardial infarction. At least, the hypoxia condition should be included in the experiment design.
- Coculture of fibroblasts with neutrophils attenuated TGF-β1 expression in activated fibroblasts as shown in Figure 4A. However, depletion of neutrophils didn't result in an elevation of TGF-β1 expression in post-infarct myocardium as shown in Figure 4C. Thus, the final conclusion needs to be further reviewed.
Author Response
Referee 1
Myocardial infarction is the most severe complication in coronary heart disease. The ischemia and subsequent reperfusion induce an innate immune response. It has been widely accepted that neutrophil induced innate immune response is closely related to fibroblast activation and polarization. However, a complicated scenario exists that inhibition of neutrophil infiltration protects the heart from acute ischemic insult but exacerbates fibrosis and heart failure. Here, Curaj et al. showed that neutrophil-fibroblast communication was important for post-infarct fibrosis via TGF-β1. There are several points to be addressed:
- To address the distinct effects of neutrophils at the acute or chronic stage of myocardial infarction, depletion prior to and after myocardial infarction is highly recommended.
Author response: We thank the reviewer for this comment. We understand the issue raised by this referee. However, since the neutrophil recruitment occurs in the first few seconds after MI, we cannot be sure that blocking neutrophil recruitment into the injured site will be achieved by applying the treatment after induction of MI. This is why we depleted neutrophils before the induction of MI. We have now provided this information in the methods section (line 115-117).
- Neutrophil depletion by antibody 1A8 didn't generate a comprehensive result. Instead, both neutrophils and monocytes showed a remarkable reduction in peripheral blood as shown in Figure 2B. Then, one of the key findings as shown in Figure 4C or 4E cannot be easily explained by neutrophil depletion.
Author response: We thank the reviewer for this comment. We have used for our study the same protocol and depletion method as described in Horckmans et al, European Heart Journal (10). They demonstrated that after treatment with 1A8 clone antibody, the recruitment of neutrophils and Ly6Chi monocytes into infarcted area was significantly reduced, while macrophage content in the heart was not influenced, or even increased (line 118-123), most probably due to the capacity of activated monocytes to proliferate. In order to show this in our model, and in addition to the blood analysis, we have now performed staining for neutrophils (anti-MPO antibody, Fig. 2C). This shows markedly reduced neutrophil infiltration 1 day after MI and for macrophages (anti-Mac3 antibody, Fig. 2D), and shows no visible differences between treated and untreated mice at later time-points after MI. The methods (line 118-123, line 163-165), figure 2 and figure legend 2 (line 137-139) have been modified accordingly.
- Neutrophils, utilized in this study, were isolated from bone marrow. Based on the mechanisms of neutrophil activation, the experiments cannot recapitulate the situation in vivo, especially after myocardial infarction. Moreover, in vitro culturing affects immune cell phenotype. Figure 1 A or B clearly show such phenotype shift during culturing. The authors should further address this issue.
Author response: We thank the reviewer for this comment. We agree with this observation. As suggested, we have now addressed this issue in our revised manuscript and have alluded to the observed phenotype shift during culturing (line 90-92).
- Monocyte-derived macrophages or cardiac resident macrophages are the important innate immune players. There is a state change from monocytes in the bone marrow to macrophage. Thus, monocytes need further activation before the coculture experiments.
Author response: We thank the reviewer for this comment. The main focus of the study was to study the role of neutrophils, therefore, after gradient separation, we have used the mononuclear fraction as control for neutrophils. Thus, it was very important to maintain the same culturing conditions as for neutrophils, to be able to have a proper control and to interpret the results in an adequate manner. To avoid further misunderstanding we have now clarified this issue in more detail in the methods section (line 86-88), and have changed monocyte fraction to mononuclear fraction throughout the manuscript.
- The results from Figure 4A are inadequate to demonstrate the effects of neutrophil-fibroblast communication on TGF-β1 production in myocardial infarction. At least, the hypoxia condition should be included in the experiment design.
Author response: We thank the reviewer for this comment. We apologize for omitting this important detail in our manuscript. We can confirm that all experiments were performed under hypoxic conditions, unless other specified. We have now corrected this issue in the revised manuscript (line 201 and line 220)
- Coculture of fibroblasts with neutrophils attenuated TGF-β1 expression in activated fibroblasts as shown in Figure 4A. However, depletion of neutrophils didn't result in an elevation of TGF-β1 expression in post-infarct myocardium as shown in Figure 4C. Thus, the final conclusion needs to be further reviewed.
Author response: We thank the reviewer for this comment. In our study, we have observed biphasic effects of neutrophils on fibroblasts:
(1) directly after MI, neutrophils induce increased TGF-ß1 production in fibroblasts.
TGF-ß1 helps to switch the pro-inflammatory phase towards an anti-inflammatory one. After neutrophil depletion, the fibroblasts do not produce TGF-ß1, and the switch to the anti-inflammatory phase is delayed.
(2) When fibroblasts differentiate towards myofibroblasts, TGF-ß1 production decreased significantly. This was independent of neutrophil depletion.
We have now clarified these findings in the revised manuscript (line 204-209, line 298-300).

Reviewer 2 Report
Curaj and colleagues hypothesized a dual action for neutrophils in myocardial infarction. In particular, they investigated the effects of neutrophils on fibroblasts and neutrophil depletion on TGF gene expression. They speculate that a reduction of TGF beta in the reparative phase after myocardial infarction would have detrimental effects in the long term. While the role of inflammation per se in many different (patho)physiological processes is indeed not a black/white phenomenon, neutrophils are likely to have distinct functions in different phases as well. However, there are several important issues that were not addressed by the authors in the current study:
Specific comments:
- By far the most important concern is the lack of any outcome data after long term neutrophil depletion in the mouse experiments. Gene expression may be interesting from a mechanistic point of view, but it is largely irrelevant if it does not lead to any functional impact such as reduced pump function, larger infarct size, infarct thinning, myocardial rupture etc.
- The methods are not adequately described: what was the timing of the in vitro experiments (how long did the authors do the co-incubation), what were the neutrophil/monocyte concentrations in those experiments, which antigen is recognized by the depleting antibody, etc.
- The same holds true for the figure legends. They are inconclusive in several cases. For example: What does the arrow in the Figure 3D insert indicate? Which part of figure 4A refers to fibroblasts and which part to myofibroblasts?
- The number of animals for the in vivo experiments is on the low side. This kind of experiments in mice usually shows large variations and a higher n would be beneficial.
- Numbers of experiments in Methods and Figures seem to deviate. Methods for example refer to n=3 replicates, but figure legends indicate n=9 experiments.
- Figure 1 only shows relative cell populations but absolute numbers need to be shown as well (i.e. cells/uL).
- The manuscript contains a number of typos and should be carefully revised accordingly.
Author Response
Referee 2
Curaj and colleagues hypothesized a dual action for neutrophils in myocardial infarction. In particular, they investigated the effects of neutrophils on fibroblasts and neutrophil depletion on TGF gene expression. They speculate that a reduction of TGF beta in the reparative phase after myocardial infarction would have detrimental effects in the long term. While the role of inflammation per se in many different (patho)physiological processes is indeed not a black/white phenomenon, neutrophils are likely to have distinct functions in different phases as well. However, there are several important issues that were not addressed by the authors in the current study:
Specific comments:
- By far the most important concern is the lack of any outcome data after long term neutrophil depletion in the mouse experiments. Gene expression may be interesting from a mechanistic point of view, but it is largely irrelevant if it does not lead to any functional impact such as reduced pump function, larger infarct size, infarct thinning, myocardial rupture etc.
Author response: We thank the reviewer for this comment. We agree that long term results would add more value to our study. However, since it has already been demonstrated that neutrophil depletion worsen heart function and increase infarct size (Horckmans et al, European Heart Journal (10)), we decided to not repeat these findings and focus on novel mechanistic insights, in order to further understand healing after MI. We have now alluded to this in the revised manuscript (line 128-131).
- The methods are not adequately described: what was the timing of the in vitro experiments (how long did the authors do the co-incubation), what were the neutrophil/monocyte concentrations in those experiments, which antigen is recognized by the depleting antibody, etc.
Author response: We thank the reviewer for this comment and apologize for this oversight. We have now provided further information on the experiments. The timing for in-vitro experiments was 24 hours, due to the phenotype shift of the neutrophils (line 90-92). Neutrophils were added 1:1 to fibroblasts, similarly to mononuclear cells (line 92). For the treatment we have used the clone 1A8 from anti-Ly6G antibody to target neutrophils (line 114-115).
- The same holds true for the figure legends. They are inconclusive in several cases. For example: What does the arrow in the Figure 3D insert indicate? Which part of figure 4A refers to fibroblasts and which part to myofibroblasts?
Author response: We thank the reviewer for this comment and apologize for this oversight. We have now added further details to figure legends. In Figure 3D, the arrow points to the identified protein which was considered for analysis (from the 2 existent selected proteins, presented as red bars). This is now indicated in the Figure legend (line 196). Further, we have added in Figure 4A the comments regarding the identification of fibroblasts (unstimulated cells -TGF-ß1) and myofibroblasts (stimulated cells +TGF-ß1) (line 202-203 and line 220).
- The number of animals for the in vivo experiments is on the low side. This kind of experiments in mice usually shows large variations and a higher n would be beneficial.
Author response: We thank the reviewer for this comment. We agree with the referee that a higher n would be more beneficial. However, in our laboratory we have found that N=5 is sufficient for the in vivo experiments given the small variability we have in the data. This is in line with new European regulations regarding animal experiments to maintain the number of animals used at minimum required for a statistical analysis. We have now alluded to this in the methods section (line 124-126).
- Numbers of experiments in Methods and Figures seem to deviate. Methods for example refer to n=3 replicates, but figure legends indicate n=9 experiments.
Author response: We thank the reviewer for this comment and apologize for this misunderstanding. We have now stated more clearly that: “Each experiment was performed in three replicates, and repeated 3 times on different days. Pooled results from all 3 days are presented (n=9).” (line 104-105 and line 144-146)
- Figure 1 only shows relative cell populations but absolute numbers need to be shown as well (i.e. cells/uL).
Author response: We thank the reviewer for this comment. We agree with the referee that an absolute number would also prove the efficiency of the treatment. Unfortunately, we have prepared and used the blood for FACS analysis and we are currently not able to provide such results. However, we have now performed additionally staining for MPO, as marker for neutrophils, to demonstrate attenuated neutrophil recruitment into the infarcted area after MI. The new results are now included in Figure 2, methods and figure legends were modified accordingly (line 118-123, line 163-165, line 137-139).
- The manuscript contains a number of typos and should be carefully revised accordingly.
Author response: We thank the reviewer for this comment and apologize for this. We have now corrected all typos and the manuscript.

Round 2
Reviewer 1 Report
Authors have explained the concerns raised in round 1.Author Response
Author response: We thank the reviewer. We apologize for not being able to perform the requested new experiments. Unfortunately, due to the current situation, our university does not allow further animal experiments for an unknown period of time. Moreover, after internal discussion, repeating already published experiments with limited scientific input does not justify the use of additional animals for the experiments. Thus, we have now introduced a limitation section, where we have commented on the small number of animals and the missing long-term experiments (line 291-295).
For a better understanding of the results, we have now reorganized the methods part and results, hoping that the manuscript will be easier to be read (Section 3.1 and 3.3 from results).
Reviewer 2 Report
The presentation of the manuscript has been considerably improved. Unfortunately, additional experiments have not been performed as requested by the reviewers.
I have no further comments.
Author Response
Author response: We apologize for not being able to perform the requested new experiments. Unfortunately, due to the current situation, our university does not allow further animal experiments for an unknown period of time. Moreover, after internal discussion, repeating already published experiments with limited scientific input does not justify the use of additional animals for the experiments. Thus, we have now introduced a limitation section, where we have commented on the small number of animals and the missing long-term experiments (line 291-295).
For a better understanding of the results, we have now reorganized the methods part and results, hoping that the manuscript will be easier to be read (Section 3.1 and 3.3 from results).